# Upper-critical solution temperature (UCST) polymer functionalized nanomedicine for controlled drug release and hypoxia alleviation in hepatocellular carcinoma therapy

**Xiaoya Niu[1,2], Yi Fu[3], Lei Feng[1,2], Maodi Xie[4], Bei Li[1,2], Lin Que[5,6], Zhen You[1,2]***

1 Department of General Surgery, Division of Biliary Surgery, West China Hospital, Sichuan University, Chengdu, 610041, China, 2 Research Center for Biliary Diseases, West China Hospital, Sichuan University, Chengdu, 610041, China, 3 Department of Physiology and Pathophysiology, School of Basic Medical Sciences, Peking University, Beijing, 100191, China, 4 West Chia-Washington Mitochondria and Metabolism Center, West China Hospital of Sichuan University, Chengdu, 610041, Sichuan, China, 5 Department of Head and Neck Oncology, West China Hospital of Stomatology, Sichuan University, Chengdu, 610041, China, 6 State Key Laboratory of Oral Diseases, West China College of Stomatology, Sichuan University, Chengdu, 610041, China

* youzhen@wchscu.cn

**Data Availability Statement:** We have uploaded the dataset to the public repository. The dataset DOI is 10.6084/m9.figshare.23367461.

## Abstract

Recently, bioinspired material such as nanoparticle has been successfully applied in the cancer therapy. However, how to precisely control the drug release from nanomedicine in tumor tissue and overcome the hypoxic microenvironment of tumor tissue is still an important challenge in the development of nanomedicine. In this work, a new type of drug-loaded nanoparticles P(AAm-co-AN)-AuNRs@CeO$_2$-DOX (PA-DOX) was prepared by combining high-efficiency photothermal reagents, critical up-conversion temperature polymer layer and anti-cancer drug doxorubicin (DOX) for the treatment of hepatocellular carcinoma (HCC). In this system, CeO$_2$ can decompose hydrogen peroxide to H$_2$O and O$_2$ alleviate the anaerobic microenvironment of liver cancer cells. As a photothermal reagent, AuNRs@-CeO$_2$ can convert near-infrared light into heat energy to achieve local heat to kill cancer cells and ablate solid tumors. In addition, the elevated temperature would enable the polymer layer to undergo a phase transition to release more DOX to achieve a controlled release mechanism, which will open up a new horizon for clinical cancer treatment.

## Introduction

As the sixth largest cancer in the world, liver cancer contributes to an important part of common tumor-related deaths [1,2]. The incidence of liver cancer has always been the focus of global health concerns. About 90% of patients with liver cancer are primary hepatocellular carcinoma. The occurrence of liver cancer is closely related to chronic liver disease, especially

**Funding:** This study was financially supported by National Natural Science Foundation' of China in the form of a grant (82272825) awarded to ZY. This study was also financially supported by Sichuan Province Science and Technology Support Program in the form of a grant (2021YFSY0038) awarded to ZY. The funders had no role in study design, data collection and analysis, decision to publish, or preparation of the manuscript.

**Competing interests:** The authors have declared that no competing interests exist.

cirrhosis. Hepatitis B virus [3], hepatitis C virus [4], excessive drinking [5] and smoking [6] are all risk factors leading to the occurrence of liver cancer. In addition, there are significant gender differences in liver cancer, with males having higher rates of new cases and mortality than females. Hepatocellular carcinoma (HCC) cells have strong aggressiveness, rapid development of the disease, and easy to metastasize to normal tissues, resulting in poor prognosis of the treatment of hepatocellular carcinoma, which poses a greater threat to the quality of life of patients.

At present, the treatment methods of HCC mainly include traditional chemical therapy, surgical resection, liver transplantation, radiotherapy, interventional therapy and new immunotherapy [7–9], among which surgical resection has a high therapeutic value in early stage patients, and liver transplantation has the highest curative ability in the treatment of middle and late stage HCC. However, patients with advanced liver cancer lack effective treatment means, and patients often rely on radiotherapy and chemotherapy to barely support the quality of life. However, in the current state of science and technology, systemic radiotherapy and chemotherapy will inevitably have some serious drawbacks, such as high cell toxicity, low tumor specificity, and poor prognosis [10].

In order to improve the outcomes of clinical treatment of cancer and the controllability and long-term efficacy of drug therapy, nanomedicine is developed rapidly as a new treatment method. Among them, nanoparticles based photothermal therapy (PTT), as one kinds of the effective non-invasive tumor therapy modalities, has received much research interest in recent years. It uses a Near Infrared (NIR) laser (wavelength 700~1400 nm) to stimulate the photothermal reagent, so that it can convert the absorbed NIR light energy into effective heat energy and kill tumor cells by thermal ablation. It has the characteristics of high selectivity, non-toxicity and easy operation [11,12]. Gold nanorods (AuNRs), as a kind of typical PTT reagent, have transverse surface plasmon resonance (TSPR) and longitudinal surface plasmon resonance (LSPR) [13]. By adjusting the aspect ratio of AuNRs, the plasmon resonance was red-shifted to the near infrared region, facilitating the PTT with deep tissue penetration [14]. In addition, compared with other shapes of gold nanomaterials, it not only has higher photothermal conversion efficiency, but also a larger photoacoustic absorption interface, resulting in stronger photoacoustic signals under the same excitation conditions. However, due to the problems of tumor heterogeneity and low photothermal conversion efficiency in deep tissue, the result of single photothermal therapy for tumor eradication is still not ideal. On the other hand, nanomaterials have also been reported to be widely used as effective drug delivery platforms for cancer therapy, including metals, hydrogels, micelles, polymers, proteins and carbon nanomaterials [15–17]. Therefore, PTT combined with nano-drug delivery systems are thought to effectively treat the tumor.

Tumor cells show a higher hypoxic level than normal cells due to external stimulation, increased metabolic activity, and mitochondrial dysfunction [18]. Hypoxia has been reported to cause resistance to chemotherapy or radiotherapy, which can significantly impair the efficacy of cancer treatment [19]. Therefore, alleviating the hypoxia is highly expected during the cancer treatment. Cerium oxide ($CeO_2$) nanoparticles have attracted increasing attention due to their catalase-like activity, which has the ability to convert $H_2O_2$ to $H_2O$ and $O_2$ to relive the hypoxia in tumor tissue to improve the efficiency of chemotherapy [20,21].

On the basis of the above considerations, in this work, we developed a drug-loaded nanoparticle P(AAm-co-AN)-AuNRs@$CeO_2$-DOX (PA-DOX) for the treatment of liver cancer. P(AAm-co-AN) as a kind of upper-critical solution temperature (UCST) polymers, has been used as drug carriers to deliver chemo-drug to tumor tissue, and its response temperature could be tuned by adjusting the feed ratio of AAm to AN [22,23].

Firstly, P(AAm-co-AN) was modified onto the surface of AuNRs@$CeO_2$. Then, the antitumor drug DOX was loaded inside the P(AAm-co-AN) modified AuNRs@$CeO_2$

nanoparticles to form the PA-DOX. The polymeric layer of P (AAm-co-AN) collapses at low temperatures and limits DOX release. The synthesis procedure of PA-DOX nanoparticles was illustrated in **S1A Striking image**. When the PA-DOX nanoparticles were injected into tumor bearing mouse, because of the enhanced permeability and retention (EPR) effect, these PA-DOX nanoparticles would accumulate in the tumor tissue. After being internalized by tumor cells, upon the irradiation by NIR 808 nm laser, the photothermal effect of AuNRs promoted the local heating and triggered the phase change of the P(AAm-co-AN) polymer, causing the polymer chain to be in a stretched state and releasing more DOX to achieve controlled long-term anti-tumor therapy. At the same time, The $CeO_2$ nanoparticles on the surface of Au NRs would catalyze the decomposition of $H_2O_2$ into $O_2$ and $H_2O$, which will alleviate the hypoxia tissue to regulate the anaerobic microenvironment of tumor tissue, potentiating the anti-tumor effect of PA-DOX (**S1B Striking image**).

## Methods and materials

### Materials

Doxorubicin hydrochloride (DOX) was purchased from J&K Scientific Technology Ltd. Co. 1,3,5-Trimethylbenzene (TMB) and acrylamide (AAm) were purchased from Macklin Reagent (Shanghai, China). Acrylonitrile (AN) and Pluronic® F-127 were purchased from Aladdin Industrial Inc (Shanghai, China). 4-Cyano-4-((phenylcarbonothioyl) thio) pentanoic acid (CPTD), 4', 6-diamidino-2-phenylindole dihydrochloride (DAPI) were obtained from Sigma-Aldrich (Missouri, USA). Annexin V-FITC Apoptosis Detection Kit was provided by KeChuang Biotechnology Inc (Suzhou, China). All other reagents and solvents were purchased from Sinopharm Chemical Reagent Inc. (Shanghai, China).

### Cells and animals

The human hepatocellular carcinoma (HCC) cell line (HepG2) were obtained from Nanjing KeyGen Biotechnology Inc (Nanjing, China). For animal studies, 48 female nude mice (weighting 18–22 g) were purchased from Qinglongshan animal breeding farm (Nanjing, China). The experiments were carried out in compliance with the Regulations of the People's Republic of China on the Administration of Laboratory Animals. The body weight and tumor size of the mice were monitored and recorded every two days.The tumor weight did not exceed 10% of the body weight of the nude mice during the tumor growth. Therefore nude mice were sacrificed using 70% carbon dioxide gas to collect tissue samples at 16 d. All animal experiments should comply with the ARRIVE guidelines. The experimental protocol was approved by the Animal Ethics Committee of West China Hospital of Sichuan University.

### Synthesis of P(AAm-co-AN)-CPTD

The UCST-type polymer P(AAm-co-AN)-CPTD was prepared via the typical reversible addition-fragmentation chain transfer (RAFT) reaction, in which CPTD was used as a chain transfer agent. First, AAm (0.726 g), AN (0.096 g), AIBN (0.0098 g) and CPTD (0.08 g) were dissolved in dioxane (4 mL), and the mixture was deoxygenated through three freeze-pump-thaw cycles under $N_2$ atmosphere. Then, the mixture was stirred at 65°C and reacted overnight for 24 h. After the reaction, the mixture was cooled down by ice water and purified by precipitation in cold methanol three times. The final product was obtained by freeze-drying, which was named as P(AAm-co-AN)-CPTD. In addition, a small portion was taken and its chemical structure was analyzed by $^1$H NMR.

## Synthesis of AuNRs

The synthesis of aunr is performed by reference to the method described earlier [24]. A typical seed-mediated growth method was used to prepare AuNRs with an aspect ratio of 4:1. In detail, $HAuCl_4$ solution (5 mL, 0.5 mM), CTAB solution (5 mL, 200 mM) and $NaBH_4$ solution (0.6 mL, 10 mM) were mixed and shaken for 2 min and incubated at 37˚C for 3 h to prepare the seed solution. Then, CTAB solution (10 mL, 200 mM), DI water (10 mL), $HAuCl_4$ solution (250 μL, 40 mM), $AgNO_3$ solution (50 μL, 40 mM), HCl (17 μL) and L-AA solution (160 μL, 100 mM) were added and mixed sequentially to form a stock solution. Afterwards, the seed solution and the stock solution were mixed in a ratio of 1:41 in a water bath at 37˚C for 24 h. Finally, the AuNRs were purified by high-speed centrifugation, dispersed in DI water, and stored under 4˚C for further use.

## Synthesis of AuNRs@CeO$_2$

To prepare AuNRs@CeO$_2$, 2 mL of AuNRs solution was diluted to 10 mL and successively added with CTAB solution (20 μL, 0.2 mM), (Hexamethylenetetramine) HMT solution (0.4 mL, 0.1M) and $Ce(NO_3)_3$ solution (0.4 mL, 0.1M). The mixture was then transferred to a 50 mL stainless steel autoclave lined with Teflon and hydrotreated at 85˚C for 5 h. Finally, AuNRs@CeO$_2$ was obtained by centrifugation and the sediment was dispersed in DI water.

## Synthesis of P(AAm-co-AN)-AuNRs@CeO$_2$-DOX (PA-DOX)

Due to the presence of sulfur atom in the P(AAM-Co-AN)-CPTD chains, P(AAM-Co-AN)-CPTD could bind AuNRs through the interaction between Au and sulfur atom. First, P(AAM-Co-AN)-CPTD (10 mg) was dissolved in a mixed solvent (2 mL, tetrahydrofuran 40: dimethylformamide 60, v/v). Under magnetic stirring, the mixture was added dropwise to the 2 mL of AuNRs@CeO$_2$ solution (10 mg/mL) through a syringe pump at a rate of 10 mL/h and kept for 24 h. Then, the mixture was purified by centrifugation to remove organic solvents and un-bound P(AAM-Co-AN)-CPTD to obtain P(AAm-co-AN)-AuNRs@CeO$_2$. The obtained sample was re-dispersed in 5 mL PBS. Then DOX (5 mg) was added and the mixture was stirred overnight in dark at 41˚C, followed by dialysis in deionized water to obtain the final product PA-DOX. The DOX loading efficiency (DLE) was 81.4% and the DOX loading capacity (DLC) of PA-DOX was 7.1%.

## Characterization

The chemical structure of P(AAm-co-AN)-CPTD was determined by nuclear magnetic resonance spectroscopy (AVANCE III HD 400, Bruker BioSpin Corp., Germany). Briefly, 2 mg of P(AAm-co-AN)-CPTD was dissolved in 1 ml of deuterated DMSO (d6-DMSO) with tetramethylsilane as an internal standard and the 1H NMR spectrum was obtained at room temperature. FT-IR Fourier transform infrared spectroscopy (FT-IR) was performed on a Bruker IFS 66V vacuum-type spectrometer through mixing lyophilized dry powder of the obtained samples with KBr and pressing it to a plate. Gel permeation chromatography (Agilent Technologies PL-GPC 50 Integrated GPC System) equipped with PLgel MIXED Columns (Agilent, particle size: 5 μm; dimensions: 7.5 mm × 300 mm) and a differential refractive index detector was applied to measure the molecular weights and distributions of the copolymers P(AAm-co-AN)-CPTD using DMSO as an eluent ($0.5$ mL·min$^{-1}$). The molecular weights of samples were calibrated with narrowly distributed dextran standards. The obtained results were shown in S1 Fig. UV-vis absorption spectrum was recorded using an ultraviolet spectrophotometer (Lengguang Technology, Shanghai). JEM-2100 JEOL Transmission Electron Microscopy (TEM)

was utilized to observe the morphology of the nanoparticles. The particle size and zeta potential were obtained using NanoBrook Omni Laser Particle Size Analyzer (Brookhaven Instruments, USA).

## The photothermal performance experiments of PA-DOX

The photothermal performance of PA-DOX was studied by an 808 nm NIR semiconductor laser device (LE-LS-808-XXTM, Leoptics, Shenzhen) with different irradiation powers. The PA-DOX PBS solution was stimulated while the real time temperature was recorded every 10 s using an electronic thermometer. PBS buffer was used as the control group. The thermal conversion efficiency of PA-DOX was calculated from the response curves. The photothermal conversion efficiency ($\eta$) of PA-DOX was calculated according to the following Formula (1).

$$\eta = [hs(T_{max,NP} - T_{surr}) - Q_{dis}]/I(1 - 10^{-A808}) \tag{1}$$

where h was the heat transfer coefficient, s was the surface area of the container, $T_{max}$ was the maximum temperature of the solution, $T_{surr}$ was the ambient temperature, I was the laser power density (3.0 W/cm$^2$), A808 was the absorption value of the material at 808 nm, and $Q_{dis}$ was the heat generated by water and the container after absorbing light. To calculate hs, the following Formulas (2) and (3) was used.

$$Q_{dis} = hs (T_{max, H_2O} - T_{surr}) \tag{2}$$

$$\tau s = (m_D - c_D)/hs \tag{3}$$

where $m_D$ is the quality of water; $c_D$ is the heat capacity of water (4.2 J/g/°C); and $\tau s$ is the time constant of the sample system, which is calculated according to the following Formulas (4) and (5).

$$t = -\tau s \ln\theta \tag{4}$$

$$\theta = (Tsurr - T)/(Tsurr - T_{max}) \tag{5}$$

## Drug loading and release

The concentration of DOX was evaluated by measuring the UV–vis absorbance of DOX at 490 nm. Firstly, that, the standard curve of the DOX concentration was set up (S3 Fig). To measure the DOX loading efficiency (DLE) and the DOX loading capacity (DLC), 5 mg of DOX was added into 10 mL of PBS with 40 mg of P(AAm-co-AN)-AuNRs@CeO$_2$. and the mixture was stirred overnight in dark at 41°C, followed by dialysis in deionized water to obtain the final product PA-DOX. The DLE and DLC were 81.4% and 9.86% respectively, calculated from Formulas (6) and (7).

$$\text{DOX loading efficiency (DLE)} = \frac{\text{DOX in Nanoparticles}}{\text{DOX Feeded}} * 100\% \tag{6}$$

$$\text{DOX loading capacity (DLC)} = \frac{\text{Weight of DOX in Nanoparticles}}{\text{Weight of Nanoparticles}} * 100\% \tag{7}$$

The effects of temperature, NIR laser and time on the release of DOX from PA-DOX were studied. Firstly, *in vitro* DOX release was carried out in 20 mL PBS buffer (pH = 7.4). 10 mg of PA-DOX in 2mL of PBS was transferred into a dialysis bag (MWCO = 3.5 kDa) and immersed in above PBS buffer. At each predetermined time point, the dialysis bag was required to be

irradiated by NIR 808 nm laser at 3.0 W/cm$^2$. Then the dialysis bag was re-immersed in PBS buffer for 15 minutes to reach a balance. After that, 3 mL of the solution was taken out from the out PBS medium and the absorption value at 490 nm was measured by ultraviolet spectrophotometer to calculate the concentration of DOX. At the same time, an equal volume of 3 mL fresh PBS buffer was refilled. The experiment was carried out in replicate.

## $H_2O_2$ decomposition and $O_2$ generation assay

In a closed system without oxygen, the decomposition of $H_2O_2$ was checked by adding 1 mL of 100 μM $H_2O_2$ into 10 mL of 200 μg/mL (Au content) AuNRs, Au@CeO$_2$ and PA-DOX respectively. In the presence or absence of NIR 808 nm laser at 3.0 W/cm$^2$, $H_2O_2$ concentration in the solution was assessed by measuring the absorbance at 405 nm every 10 min using a hydrogen peroxide determination kit. What's more, a portable dissolved oxygen meter was also applied every 10 min to monitor the $O_2$ concentration in the solution.

### *In vitro* cytotoxicity assay

The cytotoxicity of AuNRs (10, 20, 50, 100, 150, 200 μg/mL), AuNRs@CeO$_2$ (10, 20, 50, 100, 150, 200 μg/mL), free DOX (0.5, 1.0, 2.0, 4.0, 8.0, 16.0 μg/mL) and PA-DOX (0.5, 1.0, 2.0, 4.0, 8.0, 16.0 μg/mL) were evaluated by MTT assay in HepG2 cell lines. Cells were inoculated at a density of $5 \times 10^3$ cells per well in a 96-well plate and cultured at 37°C in a humidification chamber containing 5% $CO_2$ for 24 h. Then, the cell medium was replaced with fresh medium containing different samples and cells were incubated for another 48 h. Following that, the corresponding group was irradiated with NIR 808 nm laser at 3.0 W/cm$^2$ for 10 min. After irradiation, all the cells were incubated for 4 h again, and then 20 μL of MTT solution (5 mg/mL in PBS) was added to each well. This culture system was incubated for another 4 h. The medium was aspirated and 150 μL of DMSO was added to each well. After that, the plate was shaken for 10 min to fully melt the crystals. The absorbance of each well at 490 nm was determined and recorded by the microplate reader (SpectraMax Paradigm, Molecular Devices, Toronto). All cytotoxicity tests were carried out in quadruplicate.

### *In vitro* cellular migration assay

The transwell assay was used to evaluate the migration ability of HepG2 cells. Firstly, the upper chamber surface of the bottom membrane of the transwell cell was coated with Matrigel at a ratio of 1:8. The HepG2 cells incubated with medium containing AuNRs (100 μg/mL), AuNRs@CeO$_2$ (100 μg/mL), free DOX (4.0 μg/mL) and PA-DOX (4.0 μg/mL) for 48 h were digested and centrifuged with 0.25% trypsin and 0.02% EDTA, and resuspended in serum-free medium. The density of cells was adjusted to $5 \times 10^5$ cells/mL. 200 μL of cell suspension was put into the upper chamber of the Transwell cell, and 600 μL of medium containing 10% FBS was added to the lower chamber of the 24-well culture plate. After that, the culture plate was placed in a $CO_2$ incubator at 37°C for 48 h. The bottom HepG2 cells of transwell chamber was rinsed with PBS buffer twice, fix in 4% paraformaldehyde for 20 min, and then stained with crystal violet solution for 15 min. Finally, the migrating cells were viewed and imaged under the CKX53 inverted microscope (Olympus Corporation, Japan).

### *In vitro* cellular uptake

To examine the cellular uptake of free DOX and PA-DOX nanoparticles, a confocal laser scanning microscope (CLSM) imaging study was performed using HepG2 cells. Briefly, HepG2 cells were seeded on a 12-well plate with cell density at $1 \times 10^5$ cells per well and incubated for

24 h at 37˚C. Then free DOX (4.0 μg/mL) or PA-DOX (4.0 μg/mL) was added to these wells and incubated for 4 h. After incubation, the cells were rinsed 3 times with PBS buffer and fixed with 4% paraformaldehyde for 30 min at room temperature. After staining the nuclei with DAPI (200 ng/ml in PBS, Sigma) for 25 min in the dark, the stained cells were rinsed thoroughly with PBS three times to remove the free DAPI. At last, the cells were viewed and imaged under the ZX320FL confocal laser microscope (at $\lambda_{ex}$ 480 nm and $\lambda_{em}$ 590 nm, Zhongxun, Shengzhen).

### *In vivo* anti-tumour efficacy assay

HepG2 tumor-bearing mouse models were established by subcutaneously injecting HepG2 cells ($2 \times 10^6$) into the underarm of each nude mouse. After the tumor volume reached 100 mm$^3$, the nude mice were randomly divided into eight groups (saline, saline+NIR, free DOX, free DOX+NIR, AuNRs@CeO$_2$, AuNRs@CeO$_2$+NIR, PA-DOX, PA-DOX+NIR) and subjected to *in vivo* experiments. Then, the mice in each group were injected with different samples at a dose of 0.5 mg/kg DOX. After 8 h, the mice were irradiated with NIR 808 nm laser at 3.0 W/cm$^2$ at the tumor site. The body weight and tumor size of the mice were monitored and recorded every two days. Tumour volume (V) was calculated according to the formula. Tumour volume (V, mm$^3$) = $0.5 \times$ length $\times$ width$^2$. The nude mice were sacrificed on day 16 with 70% carbon dioxide gas.

### Histological assay

The mice were sacrificed on the 16[th] day, and tissues including heart, liver, spleen, lung, kidney and tumor were collected and sectioned for hematoxylin–eosin (H&E) staining. In addition, the weight of the tumor was also recorded. The H&E staining was carried out by Logone Bio, and the organs were examined and taken photos by a light microscope.

### Data analysis

Data was analyzed using GraphPad Prism 9 software and based on a single-factor analysis of variances (ANOVA) test. All data were presented as mean ± standard deviation (SD).

## Results and discussion

### Characterization of PA-DOX

In this study, UCST polymer P(AAm-co-AN)-CPTD was prepared according to the literature by the typical RAFT reaction using CPTD as the chain transfer agent [25]. The chemical structure of P(AAm-co-AN)-CPTD was analyzed by $^1$H NMR, as schematically shown in Fig 1A. The signal at 6.7~7.8 ppm revealed the presence of the amino protons units and protons in the benzene ring [26], while the signal at 1.2~1.9 ppm was attributed to the protons of -CH$_2$- on P(AAm-co-AN)-CPTD back bone [27], which proved the success synthesis of P(AAm-co-AN)-CPTD. The peaks at 3.5–3.6, and 2.0–2.5 were attributed to -CH-CONH2) and -CH-CN respectively. In view of $^1$H NMR results, the ratio of AAm to AN was nearly 68:32. The successful synthesis of P(AAm-co-AN)-CPTD was also confirmed by FT-IR measurement as shown in S1 Fig. All the characteristic absorptions of -NH$_2$ (3365 cm$^{-1}$), -CH$_2$- (2910 cm$^{-1}$), -CN (2225 cm$^{-1}$), -C = O (1690 cm$^{-1}$) were clearly seen in the spectrum. The molecular weight ($M_w$) of P(AAm-co-AN)-CPTD is 18,900, determined by gel permeation chromatography (S2 Fig). Moreover, the temperature response performance of P(AAm-co-AN)-CPTD to form the self-assembled micelles was monitored by measuring the transmittance of the solution as shown in Fig 1B. The transmittance of the polymer vesicle solution changed significantly as

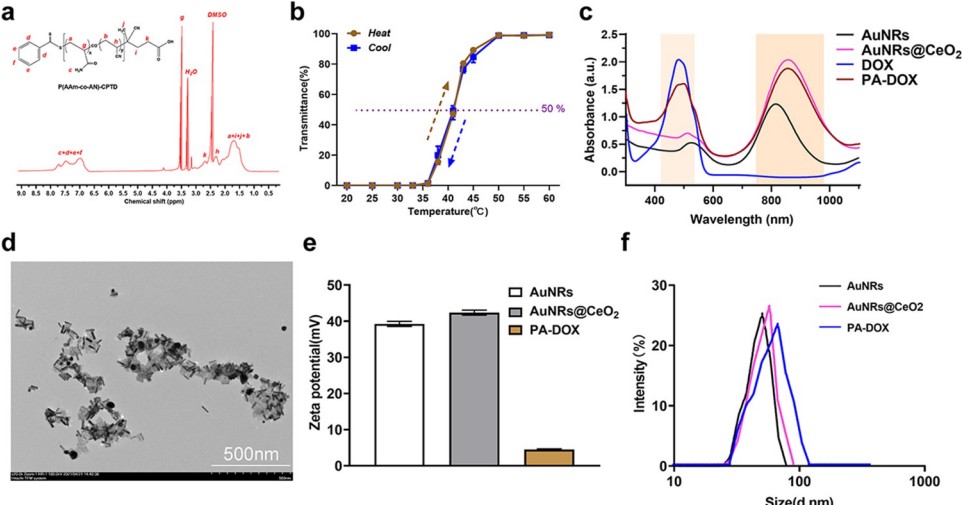

**Fig 1. Characterization of PA-DOX.** (a) $^1$H-NMR spectrum of P(AAm-co-AN)-CPTD in DMSO-$d_6$ (b) Transmittance curves of P(AAm-co-AN)-CPTD vesicles solution at different temperatures. (c) UV-Vis absorption spectra of AuNRs, AuNRs@CeO$_2$, DOX and PA-DOX, (d) TEM image of PA-DOX (inset is the image of pristine AuNRs), (e) Zeta potential of AuNRs, AuNRs@CeO$_2$ and PA-DOX, (f) DLS curves of AuNRs, AuNRs@CeO$_2$ and PA-DOX.

the temperature increased from 20˚C to 60˚C. When the test temperature was in the range of 20–36˚C, the micelle solution presented a white emulsion with a transmittance of about 0%, indicating the presence of micelles. When the temperature increased from 36˚C to 50˚C, the light transmittance of the micells solution increased from about 0% to 100%, clearly indicating the deformation of micelles. In this work, the upper critical solution temperature (UCST) of the sample was defined corresponding to the light transmittance of 50% [28,29]. It was found that the UCST of P(AAM-co-AN)-CPTD was about 42˚C, which is slightly higher than the normal body temperature. For P(AAM-co-AN)-CPTD, the interaction between acrylonitrile groups helped the polymer overcome the electrostatic repulsion, and shorten the distance between the polymer segments, thus forming the micellar aggregation and improving the probability of hydrogen bonding between the polymers. At low temperatures (< UCST), stable hydrogen bonds were formed between the polymers, allowing them to aggregate and form the micelles in the solution [30]. While at high temperatures (> UCST), the hydrogen bonds between the polymers were dissociated and the micelles were dissolved in water, resulting in the elevation of transmittance [31]. In addition, due to the different synchronization of hydrogen bonds in and between micelles, the transmittance curves did not coincide during heating and cooling procedure [32].

Fig 1C displays the UV-vis-NIR absorbance spectra of AuNRs, AuNRs@CeO$_2$, DOX and PA-DOX. The characteristic peak at 550 nm corresponded to the transverse surface plasma resonant peak of AuNRs. It can be seen that the longitudinal surface plasmon resonance (SPR) peak of AuNRs (located at 808 nm) shown a red shift to 825 nm after the coating of CeO$_2$. The full width at half maximum (FWHM) of the longitudinal and transverse SPR bands were maintained well without significant broadening, which indicated that the CeO$_2$ coating did not affected the morphology and the properties of AuNRs. In addition, except the characteristic peaks of AuNRs@CeO$_2$, the characteristic peak of Dox at 490 nm was also found in the spectrum of PA-DOX, indicating the successful loading of DOX by AuNRs@CeO$_2$. The microscopic morphology of PA-DOX was observed by TEM as shown in Fig 1D. PA-DOX exhibited the rod-like structure with a size of about 50 nm and an aspect ratio of about 4:1, similar to the

morphology of pristine AuNRs. With a close observation, we found that these PA-DOX showed a rough surface, further indicating the coating of $CeO_2$ and P(AAM-co-AN)-CPTD on their surface. Additionally, Fig 1E and 1F shows the zeta potential and size distribution of AuNRs, AuNRs@$CeO_2$ and PA-DOX. AuNRs and AuNRs@$CeO_2$ showed a high positive surface charge because of the presence of CTAB, which was used in the synthesizing procedure. After the coating of P(AAM-co-AN)-CPTD and the loading of DOX, the zeta potential of PA-DOX was greatly reduced, due to the presence of carboxyl groups in the P(AAM-co-AN)-CPTD. Furthermore, the mean diameter of AuNRs, AuNRs@$CeO_2$ and PA-DOX was 50.8, 57.3 and 69.4 nm, respectively. The gradually size increment of different samples also confirmed the deposition of $CeO_2$ and the modification of P(AAM-co-AN)-CPTD on the surface of AuNRs. Collectively, these results confirmed that PA-DOX was successfully synthesized.

## Photothermal performance

In order to evaluate the thermal conversion performance of PA-DOX nanoparticles, temperature changes of the nanoparticle solution under the NIR laser irradiation were monitored. The temperature of the solution was recorded by an electronic thermometer connected to a temperature probe. As shown in Fig 2A and 2B, the effects of NIR laser power and sample concentration on the temperature variations were studied. Firstly, the concentration of PA-DOX was fixed at 500 µg/mL, and the solution was irradiated by NIR 808 nm laser for 10 minutes with different laser power. The temperature of the solution reached as high as 50°C with the 3.0 W/$cm^2$ laser irradiation within 10 min, which was in line with the requirement of tumor ablation. In marked contrast, the temperature of the PBS buffer didn't strikingly elevate at the same condition (Fig 2B), which clearly confirmed the temperature elevation was mainly induced by the presence of PA-DOX. As we increased the irradiation power, higher solution temperature was

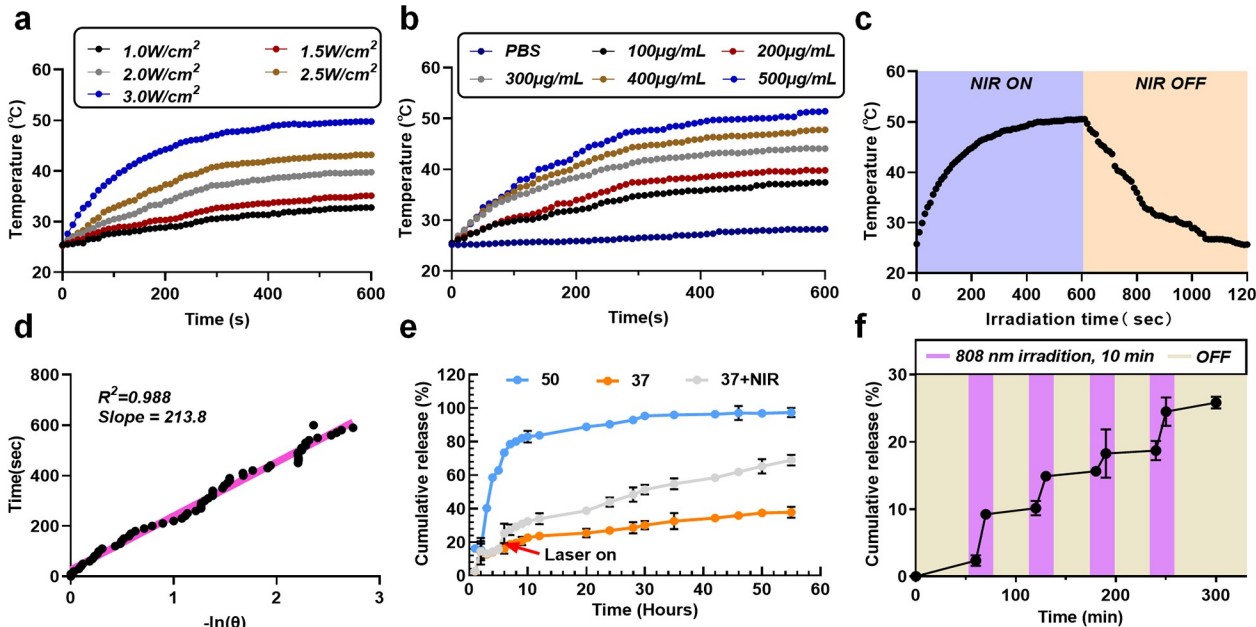

**Fig 2. Photothermal performance and Drug loading and release.** (a,b) Temperature change curves of PA-DOX solution under different powers NIR 808 nm laser irradiation, (c) Temperature change curve of PA-DOX solution after a single on/off NIR laser irradiation, (d) Relationship between the cooling time and the negative natural logarithm of temperature driving force after the NIR laser turned off, (e) Cumulative drug release curves of PA-DOX in PBS (pH 7.4) at 37°C, 50°C, and under the NIR laser irradiation, (f) Drug release curve of PA-DOX over 300 min under on/off cycles of NIR laser irradiation.

achieved by higher laser power with same irradiation time. Because the power of 3.0 W/cm$^2$ laser irradiation is high enough for tumor ablation, in the following experiment, we fixed the laser power at 3.0 W/cm$^2$ to evaluate the effect of sample concentration on the temperature variations. As shown in Fig 2B, higher sample concentration resulted in higher solution temperature with the same irradiation time. When the concentration was set at 500 μg/mL, the temperature of the solution reached 50°C within 10 min, which was feasible for photothermal therapy in cancer treatment.

To further investigate the photothermal conversion efficiency of PA-DOX, the temperature changes during the heating-cooling cycles were studied. The PA-DOX solution (500 μg/mL) was firstly irradiated by 3.0 W/cm$^2$ of 808 nm laser for 10 minutes and then the laser was turned off. The solution temperature was monitored during this procedure. According to the relationship curve between the cooling time after NIR laser turned off and the negative natural logarithm of the temperature driving force, the photothermal conversion efficiency of PA-DOX was calculated to be 34.5% (Fig 2C and 2D). These results indicate that PA-DOX exhibits a high photothermal conversion efficiency (η), which guarantees the use of PA-DOX as a PTT reagent.

## Drug loading and release

UCST polymeric layer of P(AAm-co-AN)-CPTD on the surface of PA-DOX would show an aggregating state at low temperature (<42°C) and present a dissolving or swollen state at high temperature (>42°C). When the temperature is lower than 42°C, DOX was confined inside the aggregated P(AAm-co-AN)-CPTD polymer. When the temperature is higher than 42°C, DOX will be released from the swollen polymer. So, this temperature responsible property of P(AAm-co-AN)-CPTD conferred these PA-DOX a light controlled drug release property due to the photothermal effect. therefore, the temperature sensitive releasing behavior of PA-DOX was evaluated.

Fig 2E. displays the cumulative release of DOX from PA-DOX at different temperatures with/without the NIR laser irradiation. It was found that the cumulative release of DOX from PA-DOX was temperature sensitive. When PA-DOX was maintained in the medium at 37°C, the release of DOX was very slow, and less than 40% of DOX was released from the PA-DOX even after 55 h. As the releasing temperature rose to 50°C, a fast release of DOX was achieved and about 95% of DOX was released after 30 h. This significant difference in release profiles at different temperature was due to the fact that the P(AAM-co-AN)-CPTD segment changed from a collapsed state to a stretched state when the temperature was switched from 37 to 50°C temperature [30,31], which accelerated the release of DOX.

In this work, PA-DOX exhibits a high photothermal conversion efficiency. After PA-DOX was immersed in the PBS (37°C) for 5 h, this PA-DOX solution was irradiated by 3.0 W/cm$^2$ 808 nm laser for 10 minutes. The cumulative release of DOX was also investigated as shown in Fig 2E. After the laser irradiation, the release of DOX was markedly accelerated. 1 hours post the laser irradiation, compared to the control group (PA-DOX without laser irradiation), an accelerated release of DOX from PA-DOX was clearly observed. For the control group, only 16.4% of DOX was released, while 25.4% of DOX was released from PA-DOX 1 hour post the laser irradiation. The 55-h cumulative release of DOX was about 69%, which was much higher than that released at 37°C from PA-DOX. These results confirmed that under the laser irradiation, PA-DOX achieved excellent photothermal transformation and led to local heating, which triggered the dissociation of the P(AAm-co-AN)-CPTD polymer layer and resulted in the fast release of DOX. However, the amount of DOX released from PA-DOX after laser irradiation is less than that from PA-DOX in high temperature (50°C). We think that the laser irradiation

only can heat the PA-DOX for a short time. Once the laser is turned off, the PA-DOX will slowly cool down, resulting in a decrease in the dissociation ability of the P(AAm-co-AN)-CPTD polymer layer, which consequently weakens the release of DOX. The release of DOX from PA-DOX after repeatedly laser irradiation was also tested as shown in Fig 2F. It is observed that each laser irradiation would enhance the release of DOX. Turning off the laser would suppress the DOX release. All these results demonstrated that PA-DOX could realize the controlled release of DOX upon the laser irradiation. This temperature sensitive release of DOX from PA-DOX benefits the therapy of cancer greatly. When these PA-DOX is injected into the body, the DOX would not be released out in the circulation system before PA-DOX reaches the tumor site at the body temperature. When these PA-DOX accumulates in the tumor tissue due to the Enhanced Permeability and Retention (EPR) effect, DOX will be directly released out in the tumor tissue upon the laser irradiation. So, this design of PA-DOX will enhance the anti-tumor effect of DOX while reduce the side effect of DOX against the normal tissue.

## Effect of PA-DOX on $H_2O_2$ decomposition and $O_2$ generation

The high level of hypoxia in tumor tissues makes tumor cells highly resistant to chemotherapy. Furthermore, due to the abnormal metabolism of tumor cells, the concentration of hydrogen peroxide in tumor tissues is higher than that in normal tissues. Studies have revealed that $CeO_2$ have excellent catalase and/or superoxide dismutase activities to decompose the $H_2O_2$ into $O_2$ and $H_2O$, which will alleviate the hypoxia in tumor tissue. In this work, $CeO_2$ was deposited on the surface of AuNRs to construct PA-DOX to regulate the anaerobic microenvironment of tumor tissue by virtue of the hydrogen peroxide decomposition ability of $CeO_2$. To investigate the oxygen-producing capacity of AuNRs, AuNRs@$CeO_2$, and PA-DOX, these samples were dispersed in $H_2O_2$ solution with or without NIR 808 nm laser at 3.0 W/cm². The consumption of $H_2O_2$ and the generation of $O_2$ over time were measured as shown in Fig 3A and 3B. The concentration of $H_2O_2$ decreased slowly in the AuNR solution without the laser irradiation. However, with the laser irradiation, the degradation of $H_2O_2$ became slightly fast in the AuNRs solution. We think with the laser irradiation, AuNRs will heat the solution through photothermal effect. The elevated temperature accelerates the decomposition of $H_2O_2$.

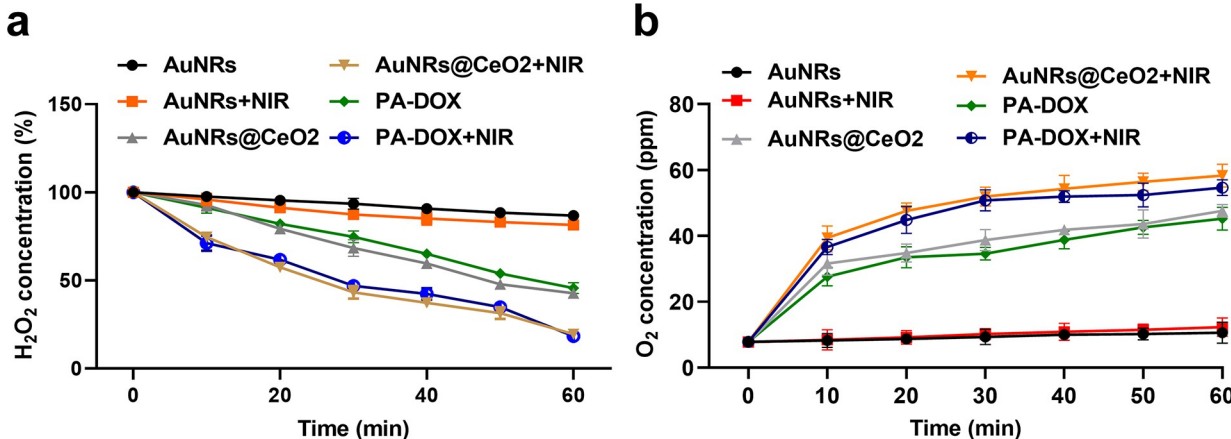

**Fig 3. Effect of PA-DOX on $H_2O_2$ decomposition and $O_2$ generation.** (a) Decomposing curves of $H_2O_2$ in the presence of AuNR, AuNRs@$CeO_2$, and PA-DOX with or without NIR 808 nm laser at 3.0 W/cm², (b) Generation curves of $O_2$ in 100 μM $H_2O_2$ solution over time under different conditions.

Compared with the AuNRs, at the presence of $CeO_2$, the decomposition of $H_2O_2$ became faster in the Au@$CeO_2$ or PA-DOX solution. Around 57.2% and 54.3% of $H_2O_2$ were degraded by Au@$CeO_2$ and PA-DOX respectively without the laser irradiation in 60 min. When irradiated by NIR laser, both PA-DOX and AuNRs@$CeO_2$ showed enhanced ability to degrade $H_2O_2$. 80.5% and 81.7% of $H_2O_2$ were decomposed by PA-DOX and AuNRs@$CeO_2$ after 60 minutes, respectively. The enhanced decomposition of $H_2O_2$ in the PA-DOX or AuNRs@$CeO_2$ solution with the laser irradiation are mostly attributed to the high temperature and the catalysis effect of $CeO_2$. The presence of P(AAm-co-AN)-CPTD on the surface PA-DOX hindered the contact between $CeO_2$ and $H_2O_2$, thus reducing the decomposing rate of $H_2O_2$ by $CeO_2$.

The concentration of $O_2$ in the solution was also measured to evaluate the generation of oxygen by PA-DOX after the laser irradiation (Fig 3B). The concentration of $O_2$ in the AuNRs solution is very low within 60 min even after the laser irradiation. This is reasonable that $H_2O_2$ is stable without the presence of $CeO_2$. However, the concentration of $O_2$ in the Au@$CeO_2$ and PA-DOX rise fast during initial 20 min and then showed a slow elevation thereafter no matter with or without the laser irradiation. These results are reasonable because $O_2$ mainly comes from the decomposition of $H_2O_2$. The presence of $CeO_2$ greatly accelerates the decomposition of $H_2O_2$, leading to a high concentration of $O_2$. At the initial stage, the high concentration of $H_2O_2$ in the solution ensured the high amount of $O_2$ generated from $H_2O_2$. Besides, the photothermal effect enhanced the catalytic ability of $CeO_2$, which also potentiated the generation of $O_2$. Collectively, the super $H_2O_2$ decomposition and $O_2$ generation abilities ensure that PA-DOX can greatly increase the $O_2$ content to alleviate the hypoxia in the tumor tissue, which facilitates the tumor therapy.

### *In vitro* cell cytotoxicity

Before the *in vivo* application of PA-DOX, the cytotoxicity of PA-DOX was tested through the MTT assay to evaluate the PTT effect on the anti-tumor ability. Firstly, the cytotoxicity of samples without the loading of DOX was evaluated. The cytotoxicity of samples with different Au concentrations is shown in Fig 4A. Without the laser irradiation, both AuNRS and AuNRS@$CeO_2$ showed a cell viability higher than 90%, even the concentration of Au reached 200 μg/mL, indicating their low cytotoxicity. The deposition of $CeO_2$ on the surface of AuNRs didn't show additional cytotoxicity against cells. When these cells were irradiated with 808 nm laser at 3.0 W/cm$^2$, their cell viabilities were greatly hampered. Even at a concentration of 50 μg/mL, AuNRS showed a reduced cell viability about 74.6%. With the coating of $CeO_2$ on the surface of AuNRS@$CeO_2$, the laser irradiation strengthened the cellular cytotoxicity and the cell viability was about 71.3%. Furthermore, the cell viability of AuNRs+NIR and AuNRS@$CeO_2$+NIR is 40.1% and 32.6% respectively when the concentration of Au is 200 μg/mL, clearly demonstrating that AuNRS@$CeO_2$ showed a slightly higher cytotoxicity than that of AuNRS at same concentration upon laser irradiation. It is reported that tumor cells would resist the photothermal therapy due to the hypoxic environment in tumor tissue. For AuNRS@$CeO_2$ nanoparticles, $CeO_2$ will decompose $H_2O_2$ to $O_2$ and $H_2O$, which consequently increases the $O_2$ content in tumor tissue thus relieving the hypoxic degree inside tumor cells. The high content of $O_2$ will sensitize the anti-tumor effect of PTT against tumor cells. So AuNRS@$CeO_2$ showed a higher anti-tumor effect against tumor cell than that of AuNRS upon laser irradiation.

Next, after the loading of DOX, MTT assay was further conducted to evaluate the antitumor effect of PA-DOX on HepG2 cells as shown in Fig 4B. All these samples showed a DOX dose dependent cytotoxicity. Except the PA-DOX+NIR, when the concentration of DOX less than

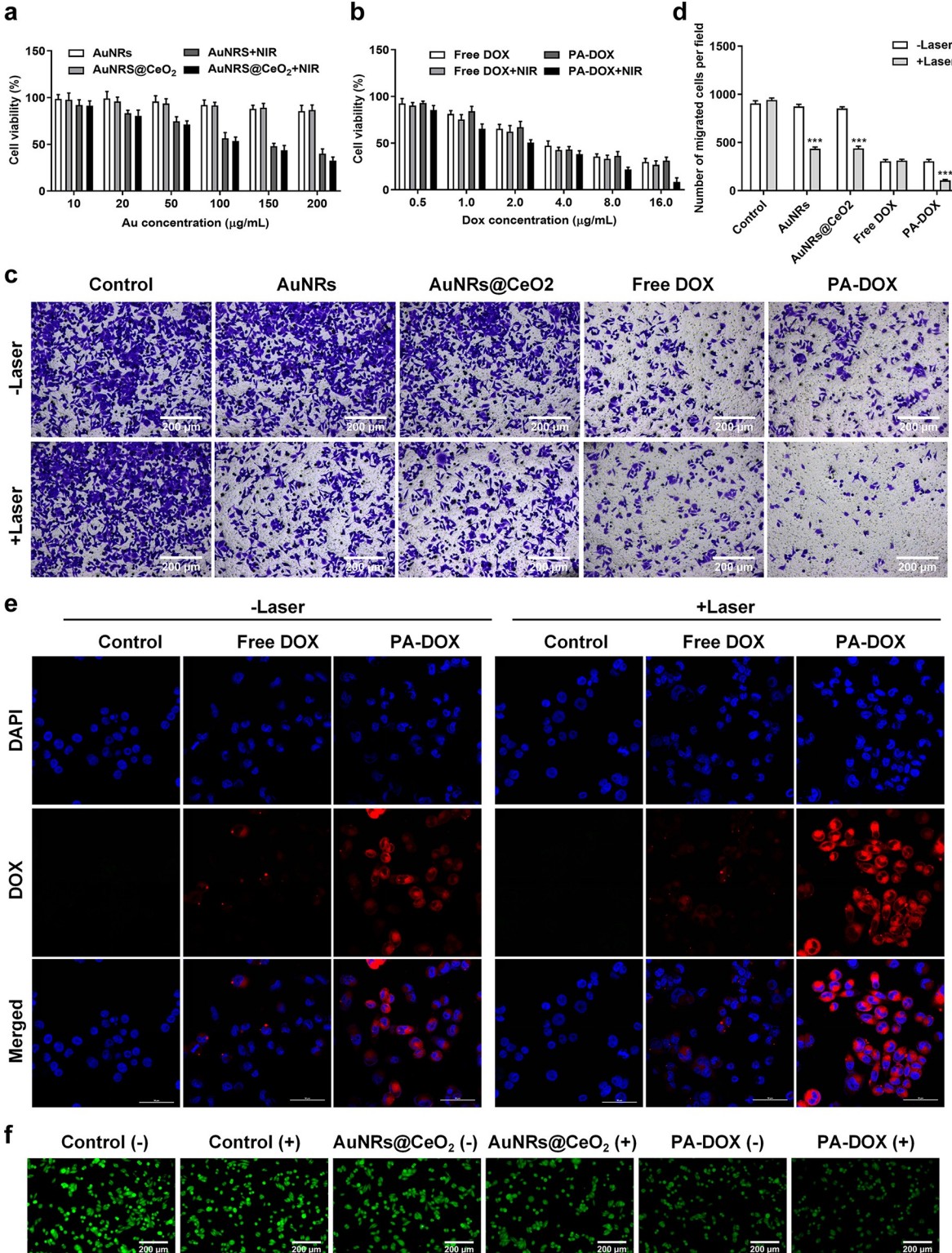

**Fig 4. Effect of PA-DOX on cell viability and migration and in vitro cellular uptake capacity.** (a, b) Cell viability of HepG2 cells treated with AuNRs, AuNRs@CeO$_2$, free DOX and PA-DOX (with/without NIR laser irradiation) at various concentrations at pH 7.4, (c) The images of cells (left to right) after treating with PBS (control), AuNRs, AuNRs@CeO$_2$, free DOX and PA-DOX (with (low row)/without (up row) NIR laser irradiation), (d) Numbers of migrating cells calculated from (c), (e) *In vitro* cellular uptake of free DOX and PA-DOX (with/without NIR laser irradiation). DOX (red), DAPI (blue), scale bar, 50 μm, (f) CLSM images of H$_2$O$_2$ in HepG2 cells treated with AuNRs (control), AuNRs@CeO$_2$, and PA-DOX (with/without NIR laser irradiation) in the presence of H$_2$O$_2$.

2 μg/mL, they showed lower cytotoxicity and the cell viability was higher than 50%. With the laser irradiation, the DOX+NIR showed slightly higher cytotoxicity than that of free DOX at the same concentration, which could be attributed to the temperature sensitive anti-tumor effect of DOX. For the PA-DOX without laser irradiation, it showed a similar cytotoxicity to free DOX at the same concentration. Although the presence of $CeO_2$ in PA-DOX would relieve the hypoxia of tumor cells to potentiate the anti-tumor effect of DOX, the retarding release of DOX from PA-DOX would reduce the effective concentration of DOX in the tumor cells which weaken its anti-tumor effect. On the contrary, PA-DOX plus NIR irradiation showed the best cytotoxicity against HepG2 cells. When the DOX concentration of PA-DOX was 2.0 μg/mL with NIR laser, the cell viability reduced to 41.6%, much lower than that of DOX +NIR group (67.8%). Even more, when the DOX concentration was 8.0 μg/mL, the cell viability of cells treated by PA-DOX was less than 20.0%. The good anti-tumor ability of PA-DOX upon the laser irradiation comes from the following reasons. (1) $CeO_2$ decomposed the $H_2O_2$ to generate the $O_2$, which relieved the hypoxia and sensitize the PTT and chemotherapy effect of PA-DOX; (2) the photothermal effect of AuNRs accelerated the release of DOX from PA-DOX upon laser irradiation due to the conformation changes of UCST polymer of P (AAm-co-AN)-CPTD, resulting in a higher effective concentration DOX inside HepG2 cells. Furthermore, the elevated temperature also enhanced the anti-tumor ability of DOX; (3) the photothermal effect of AuNRs would directly kill the tumor cells because of the high temperature. Collectively, PA-DOX showed effective anti-tumor effect against HepG2 cells under NIR laser.

### *In vitro* cellular migration assay

In order to further evaluate the antitumor effect of free DOX and PA-DOX on cell viability, the cell migration test was performed using transwell chambers. If More cells transfer from the upper wells to the lower wells, it indicates the better cell viability. Fig 4C shows the images of cancer cells in the lower wells after different treatments. It is clearly seen that the cell migration is affected by different samples. The laser irradiation also affects the cell migration greatly, and the lowest cell migration is observed in cells treated by the PA-DOX plus laser irradiation. Fig 4D shows the corresponding numerical histogram of migrated cell per well observed from Fig 4C. For the control group with the treatment of PBS, the laser irradiation didn't affect the cell migration. Compared with the control group, AuNRs and AuNRs@$CeO_2$ had no significant effect on the cell migration of HepG2 cells (Control (-): 905±28, AuNRs (-): 874±22, AuNRs@-$CeO_2$ (-): 853±18) without laser irradiation, which also confirmed that AuNRs@$CeO_2$, as the main body of drug-loaded nanoparticles, had good biocompatibility. However, free DOX and PA-DOX were found to significantly inhibited the cell migration (free DOX (-): 306±17, PA-DOX (-): 305±20) because of the toxicity of DOX. In addition, HepG2 cells treated with PA-DOX showed the lest cell migration with near-infrared light irradiation (PA-DOX(+): 107 ±9), also confirming the highest cytotoxicity of PA-DOX with the laser irradiation.

### *In vitro* cellular uptake

The cellular uptake and the localization of free DOX and PA-DOX inside tumor cells were investigated in HepG2 cells at 24 h using confocal microscopy. It can be seen from the Fig 4E that the red fluorescence of DOX and blue fluorescence of DAPI overlapped well in the group treated by free DOX after 4 h of incubation, which was attributed to the high affinity of DOX with nucleic acids. However, the red fluorescence of the free DOX group was weak inside the cells no matter with or without the lase irradiation, illustrating that few DOX could enter the cells. For PA-DOX, stronger red fluorescence of DOX was seen inside the HepG2 cells than

that in the free DOX group, which meant that PA-DOX would deliver more DOX into HepG2 cells. After NIR laser irradiation for 10 min, the strongest red fluorescence was observed in PA-DOX treated cells, indicating that NIR laser irradiation induced more DOX released from PA-DOX inside HepG2 cells. After PA-DOX was internalized into the cell through the endocytic pathway and escaped to the cytoplasm, PA-DOX was released and gradually accumulated in the nucleus to exert a cytotoxic effect. These results also explained the highest cytotoxicity of PA-DOX against HepG2 cells.

In addition, the ability of PA-DOX nanoparticles consuming the $H_2O_2$ inside tumor cells was also monitored, and the intracellular $H_2O_2$ concentration was measured as shown in Fig 4F. The $H_2O_2$ probe shows green color after its reaction with $H_2O_2$. Bright green color was observed in the control group without the laser irradiation. The intensity of green color becomes little weak after the laser irradiation, indicating that the NIR laser irradiation alone almost had no effect on the consumption of $H_2O_2$. However, when these HepG2 cells were treated with AuNRs@$CeO_2$, the intensity of green color was greatly weakened even without the laser irradiating, which supposed that $H_2O_2$ was decomposed by $CeO_2$ and there was less $H_2O_2$ reacting with the probe. For PA-DOX treated cells, the intensity of green color was also weakened. Compared with the cells treated with AuNRs@$CeO_2$, PA-DOX consume less $H_2O_2$ than that of AuNRs@$CeO_2$ at the same condition. Upon laser irradiation, the weakest intensity of green color was observed in the AuNRs@$CeO_2$ treated cells, indicating that most of the hydrogen peroxide was effectively decomposed under this condition. The PA-DOX showed comparable $H_2O_2$ decomposing ability to the AuNRs@$CeO_2$ with the laser irradiation. These results were consistent with the data from the of $H_2O_2$ decomposition experiment in Fig 3A.

## *In vivo* anti-tumor effect

In order to evaluate the *in vivo* anti-tumor activity of PA-DOX, we studied its tumor suppression effect on nude mouse model bearing liver tumor. HepG2 cells were used to establish the tumor model, and the nude mice were treated by different samples when the tumor grew to about 100 mm$^3$. All samples with or without NIR laser irradiation were studied, and PBS was used as the control. Fig 5A shows the changes of tumor volume within 14 days after the treatment. The tumor volume of PBS group continues to rise within the testing period no matter with or without the laser irradiation, and the tumor volume in the control group became 8–9 times of the initial size. For AuNRs@$CeO_2$ group without the laser irradiation, the tumor volume also continued to rise as the control group done, which partly showed the low toxicity of AuNRs@$CeO_2$. When the mice were treated with AuNRs@$CeO_2$ and laser irradiation, the growth of tumor was suppressed a little due to the combined PTT effect AuNRs and the hypoxic alleviating ability of $CeO_2$. For free DOX group, the growth of tumor was greatly retarded due to the cytotoxicity of DOX. The laser irradiation almost had no effect on the antitumor ability of DOX against the tumor bearing nude mice. Interestingly, PA-DOX showed remarkedly tumor inhibiting ability as shown in Fig 5A. During the first 6 days after the initial injection of PA-DOX, the tumor volume of the mice treated with PA-DOX without the laser irradiation became bigger slowly. After 6 days, the size of tumor volume decreased greatly. When the PA-DOX treated mice were irradiated by NIR 808 nm laser at 3.0 W/cm$^2$, the best tumor growth inhibitory effect was observed. After 14 days, a significant solid tumor ablation effect was observed in PA-DOX treated group. Fig 5B and 5C show the histogram of tumor volume and the corresponding physical photos, which also confirmed that PA-DOX with laser irradiation showed the best *in vivo* anti-tumor effect. In addition, the physiological toxicity of these samples was evaluated by monitoring the body weight changes of nude mice (Fig 5D). The PA-DOX didn't affect the body weight of the mice during the testing period. Compared

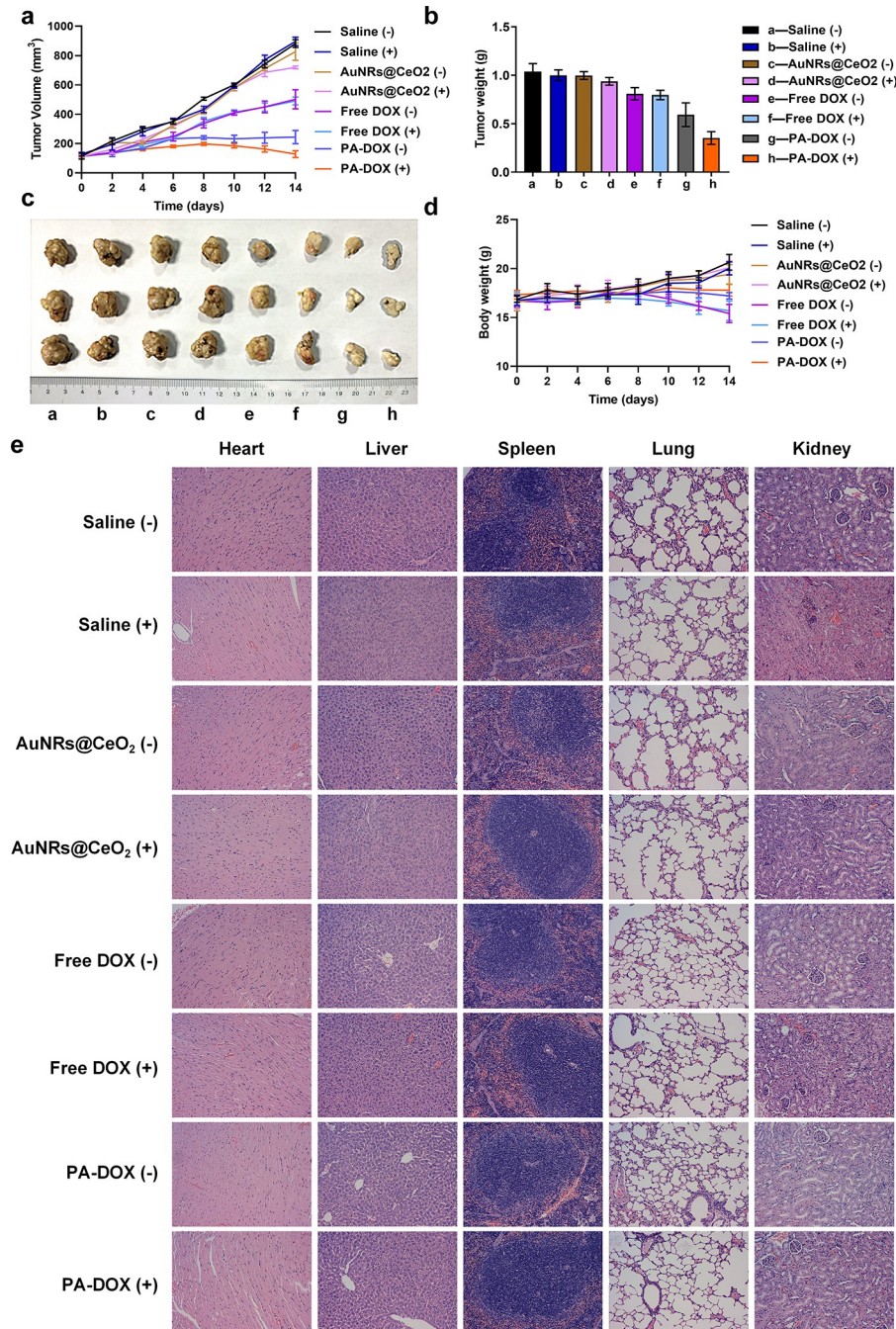

**Fig 5. Anti-tumor effect of PA-DOX *in vivo*.** (a) Tumor volume of mice treated with different samples, (b) Tumor weight of mice treated with different samples, (c) Photos of tumor tissue from mice treated with different samples, a-f were consistent with Fig 5b, (d) Body weight of mice treated with different groups, (e) H&E of tumor tissue sections from mice treated with different groups.

with the PA-DOX treated group, it could be seen that the body weight of the free DOX treated group was slightly decreased, indicating that the free DOX exhibited a acute toxicity [33]. The above results proved the high efficiency of PA-DOX to inhibit tumor growth with low cytotoxicity.

In addition, H&E staining and histological examination were performed on sections of the main organs (heart, liver, spleen, lung, and kidney) of tumor bearing nude mice (Fig 5E). The results showed that there was no obvious damage to the main organs in all the tested nude mice, indicating that the PA-DOX had low cytotoxicity and is an ideal candidate material for the treatment of liver tumors in conjunction with PTT and chemotherapy.

## Conclusions

In conclusion, we have developed a type of drug-loaded nanoparticles PA-DOX for the treatment of liver cancer. Due to the existence of AuNRs, nanoparticles had high-efficiency photothermal conversion capabilities. In addition, it had a good controllable drug release effect based on temperature changes, which provided a new idea to control the release of drug on-demand. The experimental results also show that PA-DOX can effectively decompose hydrogen peroxide to $H_2O$ and $O_2$ with the aid of $CeO_2$, which effectively alleviated the anaerobic microenvironment of tumors and strengthen the anti-tumor effect of PTT and chemotherapy. In addition, tumor borne nude mice treated with PA-DOX showed the excellent tumor suppression effect under laser irradiation. And It was found no obvious toxicity compared with free DOX-treated groups. These results indicate that the drug-loaded nanoparticles PA-DOX is a promising new material for the treatment of liver cancer.

## Supporting information

**S1 Fig. The FT-IR spectrum of P(AAm-co-AN)-CPTD.**
(PNG)

**S2 Fig. The GPC curve of P(AAm-co-AN)-CPTD, Inset presents the enlarged peak of P(AAm-co-AN)-CPTD.**
(PNG)

**S3 Fig. The Standard curve of DOX obtained from UV-Vis absorbance.**
(PNG)

**S1 Striking image. The synthesis scheme of PA-DOX NPs and their in vivo anti-tumor mechanism.**
(TIF)

## Acknowledgments

We deeply appreciate the supports by all participants.

## Author Contributions

**Conceptualization:** Xiaoya Niu, Zhen You.

**Data curation:** Yi Fu, Bei Li, Lin Que.

**Formal analysis:** Bei Li, Lin Que.

**Funding acquisition:** Zhen You.

**Investigation:** Lei Feng, Maodi Xie.

**Methodology:** Yi Fu, Lei Feng, Maodi Xie.

**Writing – original draft:** Xiaoya Niu.

**Writing – review & editing:** Xiaoya Niu, Zhen You.

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
