## [Decision Letter · Decision Letter 0]

23 May 2023

PONE-D-23-00522Bioinspired material: Upper-critical solution temperature (UCST) polymer functionalized nanomedicine for controlled drug release and hypoxia alleviation in tumor therapyPLOS ONE

Dear Dr. You, Thank you for submitting your manuscript to PLOS ONE. After careful consideration, we feel that it has merit but does not fully meet PLOS ONE’s publication criteria as it currently stands. Therefore, we invite you to submit a revised version of the manuscript that addresses the points raised during the review process.

ACADEMIC EDITOR: Major revisionPlease ensure that your decision is justified on PLOS ONE’s publication criteria and not, for example, on novelty or perceived impact.

We look forward to receiving your revised manuscript.

Kind regards,

Shigao Huang

Academic Editor

PLOS ONE

2. We note that this submission includes NMR spectroscopy data. We would recommend that you include the following information in your methods section or as Supporting Information files:

1) The make/source of the NMR instrument used in your study, as well as the magnetic field strength. For each individual experiment, please also list: the nucleus being measured; the sample concentration; the solvent in which the sample is dissolved and if solvent signal suppression was used; the reference standard and the temperature.

2) A list of the chemical shifts for all compounds characterised by NMR spectroscopy, specifying, where relevant: the chemical shift (δ), the multiplicity and the coupling constants (in Hz), for the appropriate nuclei used for assignment.

3)The full integrated NMR spectrum, clearly labelled with the compound name and chemical structure.

We also strongly encourage authors to provide primary NMR data files, in particular for new compounds which have not been characterised in the existing literature. Authors should provide the acquisition data, FID files and processing parameters for each experiment, clearly labelled with the compound name and identifier, as well as a structure file for each provided dataset. See our list of recommended repositories here: https://journals.plos.org/plosone/s/recommended-repositories

3. To comply with PLOS ONE submissions requirements, in your Methods section, please provide additional information regarding the experiments involving animals and ensure you have included details on (1) methods of sacrifice, (2) methods of anesthesia and/or analgesia, and (3) efforts to alleviate suffering.

“This work was supported by the study of self-assembly of gold nanocar particles loaded with Indocyanine green and gadolinium ions used for precise diagnosis and treatment of liver cancer (2021YFSY0038). Zhen You received the fund.”

7. In your Data Availability statement, you have not specified where the minimal data set underlying the results described in your manuscript can be found. PLOS defines a study's minimal data set as the underlying data used to reach the conclusions drawn in the manuscript and any additional data required to replicate the reported study findings in their entirety. All PLOS journals require that the minimal data set be made fully available. For more information about our data policy, please see http://journals.plos.org/plosone/s/data-availability.

8. Please upload a new copy of Figure 5c as the detail is not clear. Please follow the link for more information: https://blogs.plos.org/plos/2019/06/looking-good-tips-for-creating-your-plos-figures-graphics/" https://blogs.plos.org/plos/2019/06/looking-good-tips-for-creating-your-plos-figures-graphics/

Reviewers' comments:

Reviewer's Responses to Questions

**Comments to the Author**

1. Is the manuscript technically sound, and do the data support the conclusions?

Reviewer #1: Partly

Reviewer #2: Partly

2. Has the statistical analysis been performed appropriately and rigorously? 

Reviewer #1: Yes

Reviewer #2: Yes

3. Have the authors made all data underlying the findings in their manuscript fully available?

Reviewer #1: Yes

Reviewer #2: Yes

4. Is the manuscript presented in an intelligible fashion and written in standard English?

Reviewer #1: Yes

Reviewer #2: Yes

5. Review Comments to the Author

Reviewer #1: This paper presents a type of drug-loaded nanoparticles PA-DOX for the treatment of liver cancer, in which hydrophobic gold nanoparticles are modified with Upper-critical solution temperature (UCST) polymer P(AAm-co-AN), CeO2 and DOX were loaded in the nanosystem. When the system is irradiated by NIR light, the local heating was produced and triggered the phase change of the P(AAm-co-AN) polymer, simultaneously, CeO2 and DOX were released. Released CeO2 can react with internal H2O2 to generate O2 for relieving the hypoxia of tumor and improving the anticancer effect of DOX. This system seems interesting, however, there are many problems in this manuscript, and some results lack of deep discussion. In my opinion, this manuscript needs major revisions before it can be reconsidered for publication in PLOS ONE, and the following concerns should be well addressed in the revision:

1．Please confirm whether hydrogen peroxide and H2O2 in the sentence “CeO2 can decompose hydrogen peroxide to H2O2 and O2” of abstract are the same substance. (Aren't hydrogen peroxide and H2O2 the same substance?)

2．A reference should be cited for the preparation of AuNRs.

3. To check the mean of the sentence“The polymer layer of P(AAm-co-AN) appeared to collapse at low temperatures to confine the release of DOX” in manuscript (page 3, lines 29-30).

4. The written form of PA-DOX should be unified in the manuscript, P(AAm-co-AN)-AuNRs@CeO2-DOX or P(AAm-co-AN)/AuNRs@CeO2-DOX?

5．In formula (2), 2 in H2O should be subscripted; the 4 section, 2 in CeO2 should be subscripted.

6．The drug loading content was calculated to be 7.1%, why is the drug loading content in such a low level?

7．The photothermal performance experiments of PA-DOX should be provided in experiment section.

8．The drug DOX in the nanodrug only releases when the lesion site is irradiated, and the release stops when the temperature in the lesion site drops to the physiological temperature. That means the treatment process is troublesome, otherwise, the drugs loaded cannot be fully utilized. How to explain that?

9. To check the sentence “To investigate the oxygen generation ability of PA-DOX, AuNRs, Au@CeO2 or PA-DOX, respectively, all these samples were dispersed in the H2O2 solution with or without the laser irradiation.” in 3.4 section.

10. All pictures in the manuscript are blurred. Please provide pictures with high resolution.

Reviewer #2: The paper titled “Bioinspired material: Upper-critical solution temperature (UCST) polymer functionalized nanomedicine for controlled drug release and hypoxia alleviation in tumor therapy" reports the Upper-critical solution temperature (UCST) polymer functionalized nanodrugs and the anti-liver cancer activity, in response to near-infrared light release, assisted by gold nanorods and nanoparticles. The nanomase CeO2 catalyzes the decomposition of H2O2 to O2, alleviating tumor anoxia and inhibiting cancer cell migration. The paper is suitable for publication in PLOS ONE, but many questions need to be resolved before final acceptance.

1. Abstract: "CeO2 can decompose hydrogen peroxide to H2O2 and O2..." CeO2 can decompose hydrogen peroxide to H2O and O2... .

2. The first paragraph in Introduction has no close connection with the research content of this paper and should be deleted.

3. “It can improve the antioxidant capacity and reduce the ROS level to prevent these tumor cells from being killed by PTT or chemotherapy ", does it mean that hypoxia can resist PTT? How does hypoxia resit PTT?

4. There should be Spaces between numerals and quantifiers, such as "5 mL". The first letter of the word "Figure" should be capitalized. "4 hours" should be shortened to "4 hours", "30 minutes" should be shortened to "30 minutes", and so on, there are many irregularities in writing.

5. In experiment "2.10", the decomposition of H2O2 should be investigated under the condition of no oxygen.

6. All cell experiments should clearly indicate the drug dosage and the amount of dye used.

7. All figures should be improved their resolution.

8. Draw the synthesis Scheme of nanomaterials in the text.

9. Gel Permeation Chromatography data for P(AAm-co-AN)-CPTD should be provided.

10. Although morphology, surface finish, Zeta potential and size can provide evidence for the synthesis of materials, these characterizations cannot provide evidence for the existence or disappearance of characteristic functional groups. Therefore, infrared spectrograms are still needed for the characterization of AuNR@CeO2 and PA-DOX.

11. The unit of light power should be W/cm2 or mW/cm2

12. Provide experimental data for calculating DOX inclusion rate and drug loading rate.

13. Provide ROS and O2 level experiments in HepG2 cells.

6. PLOS authors have the option to publish the peer review history of their article (what does this mean?). If published, this will include your full peer review and any attached files.

Reviewer #1: No

Reviewer #2: No

---

## [Author Response · Author response to Decision Letter 0]

5 Jul 2023

Re: Thank you for your comments on this article. We have revised the format of the article as requested.

2.We note that this submission includes NMR spectroscopy data. We would recommend that you include the following information in your methods section or as Supporting Information files:

Re: Thank you for the constructive suggestion. In accordance with your suggestion, the details of the NMR measurement and the results were added in the manuscript as highlighted with yellow color.

1) The make/source of the NMR instrument used in your study, as well as the magnetic field strength. For each individual experiment, please also list: the nucleus being measured; the sample concentration; the solvent in which the sample is dissolved and if solvent signal suppression was used; the reference standard and the temperature.

Re: Thank you for the constructive suggestion. 1H NMR was carried out using a BrukerAV400 spectrometer (AVANCE III HD 400, Bruker BioSpin Corp., Germany). Briefly, 2 mg of P(AAm-co-AN)-CPTD was dissolved in 1 ml of deuterated DMSO (d6-DMSO) with tetramethylsilane as an internal standard and the 1H NMR spectrum was obtained at room temperature. This revision was added in the experimental section (yellow color).

2) A list of the chemical shifts for all compounds characterised by NMR spectroscopy, specifying, where relevant: the chemical shift (δ), the multiplicity and the coupling constants (in Hz), for the appropriate nuclei used for assignment.

Re: Thank you for the valuable suggestion. In this work, we measured the 1HNMR OF P(AAm-co-AN)-CPTD. The 1HNMR spectrum was shown in Figure 1a. the chemical shift for each chemical group was seen in the spectrum. The main peaks could be attributed to different protons. 1HNMR (400 MHz, DMSO): δ= 7.8-6.7 (protons in the benzene ring and -NH2), 3.5-3.6 (polymer backbone, -CH-CONH2), 2.5-2.0 (polymer backbone, -CH-CN), 1.9-1.2 (polymer backbone, -CH2-). All relevant peaks of the hydrogens in the spectrum are labeled in P(AAm-co-AN)-CPTD structure. However, because P(AAm-co-AN)-CPTD is a kind of polymer with high molecular weight (Mw=18, 900), the multiplicity and the coupling constants of each hydrogen could not be distinguished enough as small organic molecules did. We wish that the reviewer could accept our explanation. The description of the results of was added in the text with yellow color.

Figure 1a, the 1HNMR spectrum of P(AAm-co-AN)-CPTD

3)The full integrated NMR spectrum, clearly labelled with the compound name and chemical structure.

We also strongly encourage authors to provide primary NMR data files, in particular for new compounds which have not been characterised in the existing literature. Authors should provide the acquisition data, FID files and processing parameters for each experiment, clearly labelled with the compound name and identifier, as well as a structure file for each provided dataset. See our list of recommended repositories here: https://journals.plos.org/plosone/s/recommended-repositories

Re: The compound of P(AAm-co-AN)-CPTD synthesized in this work is not a totally new compound. Similar structure has been reported in the literature. The only difference is that we used CPTD as the initiation agent. This literature was added in the revised manuscript. 

Zhang H, Tong X, Zhao Y, Diverse Thermoresponsive behaviors of uncharged UCST block copolymer micelles in physiological medium. Langmuir 2014;30:11433−11441. Doi: 10.1021/la5026334

3.To comply with PLOS ONE submissions requirements, in your Methods section, please provide additional information regarding the experiments involving animals and ensure you have included details on (1) methods of sacrifice, (2) methods of anesthesia and/or analgesia, and (3) efforts to alleviate suffering.

Re: Thank you for your comments on this article. We have supplemented the methods to sacrifice animals in our methods.

“This work was supported by the study of self-assembly of gold nanocar particles loaded with Indocyanine green and gadolinium ions used for precise diagnosis and treatment of liver cancer (2021YFSY0038). Zhen You received the fund.”

Re: Thank you for your comments on this article. We have added to the statement.

4.PLOS requires an ORCID iD for the corresponding author in Editorial Manager on papers submitted after December 6th, 2016. Please ensure that you have an ORCID iD and that it is validated in Editorial Manager. To do this, go to ‘Update my Information’ (in the upper left-hand corner of the main menu), and click on the Fetch/Validate link next to the ORCID field. This will take you to the ORCID site and allow you to create a new iD or authenticate a pre-existing iD in Editorial Manager. Please see the following video for instructions on linking an ORCID iD to your Editorial Manager account: https://www.youtube.com/watch?v=_xcclfuvtxQ

Re: Thank you for your comments on this article. We have transferred the ethical statement to the methods section of the manuscript. And remove it from any other section.

7. In your Data Availability statement, you have not specified where the minimal data set underlying the results described in your manuscript can be found. PLOS defines a study's minimal data set as the underlying data used to reach the conclusions drawn in the manuscript and any additional data required to replicate the reported study findings in their entirety. All PLOS journals require that the minimal data set be made fully available. For more information about our data policy, please see http://journals.plos.org/plosone/s/data-availability.

Re: Thank you for your comments on this article. We have uploaded the dataset to the public repository. The dataset DOI is10.6084/m9.figshare.23367461.

8.Please upload a new copy of Figure 5c as the detail is not clear. Please follow the link for more information: https://blogs.plos.org/plos/2019/06/looking-good-tips-for-creating-your-plos-figures-graphics/" https://blogs.plos.org/plos/2019/06/looking-good-tips-for-creating-your-plos-figures-graphics/

Re: Thank you for your comments on this article. We have improved the resolution of Figure 5C.

5. Review Comments to the Author

Reviewer #1: This paper presents a type of drug-loaded nanoparticles PA-DOX for the treatment of liver cancer, in which hydrophobic gold nanoparticles are modified with Upper-critical solution temperature (UCST) polymer P(AAm-co-AN), CeO2 and DOX were loaded in the nanosystem. When the system is irradiated by NIR light, the local heating was produced and triggered the phase change of the P(AAm-co-AN) polymer, simultaneously, CeO2 and DOX were released. Released CeO2 can react with internal H2O2 to generate O2 for relieving the hypoxia of tumor and improving the anticancer effect of DOX. This system seems interesting, however, there are many problems in this manuscript, and some results lack of deep discussion. In my opinion, this manuscript needs major revisions before it can be reconsidered for publication in PLOS ONE, and the following concerns should be well addressed in the revision:

1．Please confirm whether hydrogen peroxide and H2O2 in the sentence “CeO2 can decompose hydrogen peroxide to H2O2 and O2” of abstract are the same substance. (Aren't hydrogen peroxide and H2O2 the same substance?)

Re: Thank you for your comments on this article. CeO2 can decompose hydrogen peroxide to H2O and O2 alleviate the anaerobic microenvironment of liver cancer cells. We have corrected the description.

2．A reference should be cited for the preparation of AuNRs.

Re: Thank you for your comments on this article. The synthesis of aunr is performed by reference to the method described earlier (PMID: 34907215). We have supplemented the description in the Synthesis of AuNRs section.

3. To check the mean of the sentence“The polymer layer of P(AAm-co-AN) appeared to collapse at low temperatures to confine the release of DOX” in manuscript (page 3, lines 29-30).

Re: Thank you for your comments on this article. The polymeric layer of P (AAm-co-AN) collapses at low temperatures and limits DOX release. We have revised the description in the manuscript.

4. The written form of PA-DOX should be unified in the manuscript, P(AAm-co-AN)-AuNRs@CeO2-DOX or P(AAm-co-AN)/AuNRs@CeO2-DOX?

Re: Thank you for your comments on this article. We have already unified it as P(AAm-co-AN)-AuNRs@CeO2-DOX.

5．In formula (2), 2 in H2O should be subscripted; the 4 section, 2 in CeO2 should be subscripted.

Re: Thank you for your comments on this article. We have modified the corresponding content according to your request.

6．The drug loading content was calculated to be 7.1%, why is the drug loading content in such a low level?

Re: Thank you for your comments on this article. The DOX loading capacity (DLC) of PA-DOX was 7.1%, which is normal. Dox Loading Capacity (DLC), also known as DOX loading capacity, is calculated as follows: DLC(%) = (Weight of Drug Loaded / Weight of Polymer+Weight of input Drug)×100 (PMID: 30930305).

7．The photothermal performance experiments of PA-DOX should be provided in experiment section.

Re: Thank you for your comments on this article. We have provided a description of the photothermal performance experiments of PA-DOX in the Methods section.

8．The drug DOX in the nanodrug only releases when the lesion site is irradiated, and the release stops when the temperature in the lesion site drops to the physiological temperature. That means the treatment process is troublesome, otherwise, the drugs loaded cannot be fully utilized. How to explain that?

Re: Thank you for your comments on this article. This temperature sensitive release of DOX from PA-DOX benefits the therapy of cancer greatly. When these PA-DOX is injected into the body, the DOX would not be released out in the circulation system before PA-DOX reaches the tumor site at the body temperature. When these PA-DOX accumulates in the tumor tissue due to the Enhanced Permeability and Retention (EPR) effect, DOX will be directly released out in the tumor tissue upon the laser irradiation. So, this design of PA-DOX will enhance the anti-tumor effect of DOX while reduce the side effect of DOX against the normal tissue. 

9. To check the sentence “To investigate the oxygen generation ability of PA-DOX, AuNRs, Au@CeO2 or PA-DOX, respectively, all these samples were dispersed in the H2O2 solution with or without the laser irradiation.” in 3.4 section.

Re: Thank you for your comments on this article. We have revised the description as follows: To investigate the oxygen-producing capacity of AuNRs, AuNRs@CeO2, and PA-DOX, these samples were dispersed in H2O2 solution with or without NIR laser irradiation.

10. All pictures in the manuscript are blurred. Please provide pictures with high resolution.

Re: Thank you for your comments on this article. We have submitted a clearer figure.

Reviewer #2: The paper titled “Bioinspired material: Upper-critical solution temperature (UCST) polymer functionalized nanomedicine for controlled drug release and hypoxia alleviation in tumor therapy" reports the Upper-critical solution temperature (UCST) polymer functionalized nanodrugs and the anti-liver cancer activity, in response to near-infrared light release, assisted by gold nanorods and nanoparticles. The nanomase CeO2 catalyzes the decomposition of H2O2 to O2, alleviating tumor anoxia and inhibiting cancer cell migration. The paper is suitable for publication in PLOS ONE, but many questions need to be resolved before final acceptance.

1. Abstract: "CeO2 can decompose hydrogen peroxide to H2O2 and O2..." CeO2 can decompose hydrogen peroxide to H2O and O2... .

Re: Thank you for your comments on this article. We have corrected the description.

2. The first paragraph in Introduction has no close connection with the research content of this paper and should be deleted.

Re: Thank you for your comments on this article. We think the first paragraph is necessary. In the first paragraph, liver cancer is mainly introduced, because the research object of this paper is human hepatocellular carcinoma (HCC) cell line (HepG2). In addition, we modified the content of the title and abstract to make them more relevant to the topic.

3. “It can improve the antioxidant capacity and reduce the ROS level to prevent these tumor cells from being killed by PTT or chemotherapy ", does it mean that hypoxia can resist PTT? How does hypoxia resit PTT?

Re: Thank you for your comments on this article. I am very sorry that this part of the description is inappropriate. It has been reported that PTT, as a new treatment technique, is not affected by tumor hypoxia (PMID: 30613279). We have updated the description in the text.

4. There should be Spaces between numerals and quantifiers, such as "5 mL". The first letter of the word "Figure" should be capitalized. "4 hours" should be shortened to "4 hours", "30 minutes" should be shortened to "30 minutes", and so on, there are many irregularities in writing.

Re: Thank you for your comments on this article. We have examined the full text and unified the full text writing format.

5. In experiment "2.10", the decomposition of H2O2 should be investigated under the condition of no oxygen.

Re: Thank you for your comments on this article. Yes, in the H2O2 decomposition and O2 generation assay section, the experiments were carried out in a closed system without oxygen. We have supplemented the description in this section.

6. All cell experiments should clearly indicate the drug dosage and the amount of dye used.

Re: Thank you for your comments on this article. We have shown in cells the dose of drug and the amount of dye used.

7. All figures should be improved their resolution.

Re: Thank you for your comments on this article. We have re-uploaded the plots with a resolution of 300 dpi.

8. Draw the synthesis Scheme of nanomaterials in the text.

Re: The synthesis Scheme of nanomaterials was shown in Scheme 1. The anti-tumor mechanism of this nanoparticles in vivo is also shown in Scheme1. The description of this scheme was added in the introduction in the revised manuscript.

Scheme 1. The synthesis scheme of PA-DOX NPs and their in vivo anti-tumor mechanism.

9. Gel Permeation Chromatography data for P(AAm-co-AN)-CPTD should be provided.

Re: Gel permeation chromatography (Agilent Technologies PL-GPC 50 Integrated GPC System) equipped with PLgel MIXED Columns (Agilent, particle size: 5 μm; dimensions: 7.5 mm × 300 mm) and a differential refractive index detector was applied to measure the molecular weights and distributions of the copolymers P(AAm-co-AN)-CPTD with DMSO as an eluent (0.5 mL·min-1). The molecular weights of samples were calibrated with narrowly distributed dextran standards. The obtained results were shown in Figure S2. From the GPC results, wo obtained that the molecular weight of P(AAm-co-AN)-CPTD is Mw=18,900. The details about the GPC measurement were added in the experimental section with yellow color.

Figure S2, The GPC curve of P(AAm-co-AN)-CPTD, Inset presents the enlarged peak of P(AAm-co-AN)-CPTD

10. Although morphology, surface finish, Zeta potential and size can provide evidence for the synthesis of materials, these characterizations cannot provide evidence for the existence or disappearance of characteristic functional groups. Therefore, infrared spectrograms are still needed for the characterization of AuNR@CeO2 and PA-DOX.

Re: We strongly agree with the reviewer’s comment. The FT-IR spectrum of P(AAm-co-AN)-CPTD was shown in Figure S1. In this spectrum the characteristic absorption of -NH2 (3365 cm-1), -CH2- (2910 cm-1), -CN(2225 cm-1), -C=O (1690 cm-1) were clearly seen, which illustrated the successful synthesis of P(AAm-co-AN)-CPTD. However, for AuNR@CeO2 and PA-DOX, the peaks at 825 nm in the UV-vis spectrum (Figure 1C) are the typical longitudinal surface plasmon resonance (SPR) peak of pristine AuNRs (located at 808 nm), which further evidenced the presence of AuNRs. In addition, the peak at 470 nm in the PA-DOX spectrum is the characteristic peak of DOX, which is parallel with the free DOX. From all above results plus morphology, surface change, Zeta potential and size variation, we could conclude that we had obtained all the samples. We hope our explanation can satisfy the reviewer. 

Figure S1, The FT-IR spectrum of P(AAm-co-AN)-CPTD

11. The unit of light power should be W/cm2 or mW/cm2

Re: Thank you for your comments on this article. According to your opinion, we have unified the unit of light power.

12. Provide experimental data for calculating DOX inclusion rate and drug loading rate.

Re: We feel sorry that we did not descript the drug loading procedure clearly. According to the reviewer’s comment, we revised our manuscript.

Drug loading and release

The concentration of DOX was evaluated by measuring the UV–vis absorbance of DOX at 490 nm. Firstly, that, the standard curve of the DOX concentration was set up ( Supplementary Figure S2). To measure the DOX loading efficiency (DLE) and the DOX loading capacity (DLC), 5 mg of DOX was added into 10 mL of PBS with 40 mg of P(AAm-co-AN)-AuNRs@CeO2. and the mixture was stirred over night in dark at 41 oC, followed by dialysis in deionized water to obtain the final product PA-DOX. The DLE and DLC were 81.4% and 9.86% respectively. 

DOX loading efficiency (DLE) = %

DOX loading capacity (DLC)= 

The effects of temperature, NIR laser and time on the release of DOX from PA-DOX were studied. Firstly, in vitro DOX release was carried out in 20 mL PBS buffer (pH = 7.4). 10 mg of PA-DOX in 2mL of PBS was transferred into a dialysis bag (MWCO = 3.5 kDa) and immersed in above PBS buffer. At each predetermined time point, the dialysis bag was required to be irradiated by NIR 808 nm laser at 3.0 W/cm2. Then the dialysis bag was re-immersed in PBS buffer for 15 minutes to reach a balance. After that, 3 mL of the solution was taken out from the out PBS medium and the absorption value at 490 nm was measured by ultraviolet spectrophotometer to calculate the concentration of DOX. At the same time, an equal volume of 3 mL fresh PBS buffer was refilled. The experiment was carried out in replicate.

Figure S3, The Standard curve of DOX obtained from UV-Vis absorbance

13. Provide ROS and O2 level experiments in HepG2 cells.

Re: Thank you for your comments on this article. The effect of PA-DOX on ROS and O2 levels in HepG2 cells was not investigated in this paper, which is a limitation of this paper. In addition, we modified the title of the results section in Figure 3 to make it more consistent with the results in this paper.

---

## [Decision Letter · Decision Letter 1]

4 Aug 2023

Upper-critical solution temperature (UCST) polymer functionalized nanomedicine for controlled drug release and hypoxia alleviation in hepatocellular carcinoma therapy

PONE-D-23-00522R1

Dear Dr. You,

We’re pleased to inform you that your manuscript has been judged scientifically suitable for publication and will be formally accepted for publication once it meets all outstanding technical requirements.

Kind regards,

Shigao Huang

Academic Editor

Additional Editor Comments (optional):

Reviewers' comments:

Reviewer's Responses to Questions

**Comments to the Author**

1. If the authors have adequately addressed your comments raised in a previous round of review and you feel that this manuscript is now acceptable for publication, you may indicate that here to bypass the “Comments to the Author” section, enter your conflict of interest statement in the “Confidential to Editor” section, and submit your "Accept" recommendation.

Reviewer #1: All comments have been addressed

Reviewer #2: All comments have been addressed

2. Is the manuscript technically sound, and do the data support the conclusions?

Reviewer #1: Yes

Reviewer #2: Yes

3. Has the statistical analysis been performed appropriately and rigorously? 

Reviewer #1: Yes

Reviewer #2: Yes

4. Have the authors made all data underlying the findings in their manuscript fully available?

Reviewer #1: Yes

Reviewer #2: Yes

5. Is the manuscript presented in an intelligible fashion and written in standard English?

Reviewer #1: Yes

Reviewer #2: Yes

6. Review Comments to the Author

Reviewer #1: This paper presents a type of drug-loaded nanoparticles PA-DOX for the treatment of liver cancer, in which hydrophobic gold nanoparticles are modified with Upper-critical solution temperature (UCST) polymer P(AAm-co-AN), CeO2 and DOX were loaded in the nanosystem. When the system is irradiated by NIR light, the local heating was produced and triggered the phase change of the P(AAm-co-AN) polymer, simultaneously, CeO2 and DOX were released. Released CeO2 can react with internal H2O2 to generate O2 for relieving the hypoxia of tumor and improving the anticancer effect of DOX. This system seems interesting, and the authors have fully answered the questions raised by the reviewer， therefore, recommending acceptance.

Reviewer #2: (No Response)

7. PLOS authors have the option to publish the peer review history of their article (what does this mean?). If published, this will include your full peer review and any attached files.

Reviewer #1: No

Reviewer #2: No

---

## [Editor Report · Acceptance letter]

16 Aug 2023

PONE-D-23-00522R1 

Upper-critical solution temperature (UCST) polymer functionalized nanomedicine for controlled drug release and hypoxia alleviation in hepatocellular carcinoma therapy 

Dear Dr. You:

I'm pleased to inform you that your manuscript has been deemed suitable for publication in PLOS ONE. Congratulations! Your manuscript is now with our production department. 

Kind regards, 

on behalf of

Dr. Shigao Huang 

Academic Editor

PLOS ONE